

# Unique haplotypes of cacao trees as revealed by *trnH-psbA* chloroplast DNA

Nidia Gutiérrez-López[1], Isidro Ovando-Medina[1], Miguel Salvador-Figueroa[1], Francisco Molina-Freaner[2], Carlos H. Avendaño-Arrazate[3] and Alfredo Vázquez-Ovando[1]

[1] Instituto de Biociencias, Universidad Autónoma de Chiapas, Tapachula, Chiapas, Mexico

[2] Departamento de Ecología de la Biodiversidad, Instituto de Ecología, Universidad Nacional Autónoma de México, Hermosillo, Sonora, Mexico

[3] Campo Experimental Rosario Izapa, Instituto Nacional de Investigaciones Forestales, Agrícolas y Pecuarias, Tuxtla Chico, Chiapas, Mexico

Corresponding author
Alfredo Vázquez-Ovando,
jose.vazquez@unach.mx

## ABSTRACT

Cacao trees have been cultivated in Mesoamerica for at least 4,000 years. In this study, we analyzed sequence variation in the chloroplast DNA *trnH-psbA* intergenic spacer from 28 cacao trees from different farms in the Soconusco region in southern Mexico. Genetic relationships were established by two analysis approaches based on geographic origin (five populations) and genetic origin (based on a previous study). We identified six polymorphic sites, including five insertion/deletion (indels) types and one transversion. The overall nucleotide diversity was low for both approaches (geographic = 0.0032 and genetic = 0.0038). Conversely, we obtained moderate to high haplotype diversity (0.66 and 0.80) with 10 and 12 haplotypes, respectively. The common haplotype (H1) for both networks included cacao trees from all geographic locations (geographic approach) and four genetic groups (genetic approach). This common haplotype (ancient) derived a set of intermediate haplotypes and singletons interconnected by one or two mutational steps, which suggested directional selection and event purification from the expansion of narrow populations. Cacao trees from Soconusco region were grouped into one cluster without any evidence of subclustering based on AMOVA ($F_{ST} = 0$) and SAMOVA ($F_{ST} = 0.04393$) results. One population (Mazatán) showed a high haplotype frequency; thus, this population could be considered an important reservoir of genetic material. The indels located in the *trnH-psbA* intergenic spacer of cacao trees could be useful as markers for the development of DNA barcoding.

## INTRODUCTION

The Neotropical cacao tree (*Theobroma cacao* L.) is widely cultivated in Central and South America, Africa, Indonesia and Malaysia. It is considered an economically important crop because its seeds are used in the chocolate industry (*Wood, 2001*). Trees can be traditionally classified based on agromorphological traits as Criollo, Forastero and Trinitario (*Cheesman, 1944*; *Toxopeus, 1985*). In Mesoamerica, the Criollo cacao has been widely used as food and other purposes as well for nearly 4,000 years (*De la Cruz et al., 1995*; *Whitkus et al., 1998*; *Powis et al., 2011*).
*Motamayor et al. (2008)* proposed 10 cacao genetic groups based on simple sequence repeat (SSR) analysis. Under this genetic classification, only the traditional group of Criollo has been retained as an accepted genetic group. Forastero contains members of the other nine genetic groups from South America (*Motamayor et al., 2008*) while Trinitarios are believed to be hybrids of various groups (*Yang et al., 2013*). South America has been reported to contain the highest genetic diversity of cacao trees.

Conversely, the genetic diversity of cacaos in southern Mexico was reported to be moderate to low in natural populations (*Whitkus et al., 1998* using RAPD markers) and cultivated forms (*Vázquez-Ovando et al., 2014* using microsatellite markers), although a wide diversity in cacao pod (fruit smooth, rugose, very rugose; apex blunt, apex point; cylindrical and oblate form; basal constriction slight/absent to very pronounced; colours green, red), and seed (elliptics and oblate forms; colour white, slightly pigment and purple intense) morphologies was observed and reported by *Avendaño-Arrazate et al. (2010)*. In the Soconusco farms (Chiapas, Mexico), *Vázquez-Ovando et al. (2014)* found moderate to high allelic richness and high levels of homozygosity. The authors reported the presence of trees sharing genetic identity with those considered "Ancient Criollo" but also reported the presence of private alleles. These alleles may be associated with commercially relevant phenotypic traits that preserve their relationship with other polymorphic regions of the DNA.

The chloroplast DNA (cpDNA) and its markers have been increasingly used for studies of genetic population structure, evolution, gene flow, haplotype frequency and phylogenetic relationships. Due to its high conservation due to maternal uniparental inheritance, cpDNA is the main data source used for the construction of phylogenetic relationships in plants (*Shaw & Small, 2005*). In addition, the cpDNA contains variable DNA regions, which makes them useful for studies of population genetics and conservation issues (*Shaw & Small, 2005*; *Shaw et al., 2007*). These regions have been widely used to establish phylogeography patterns in alpine species (*Wang et al., 2008*), to gain insight into the center of origin of cultivated grape populations in Europe (*Arroyo-García et al., 2006*) and to explain the diversity and population structure of cultivated Chinese cherries (*Chen et al., 2013*).

Although cpDNA has not been commonly used in cocoa studies, the technique was employed to analyze population genetic variability and to elucidate the complex origins of cocoa varieties. *Yang et al. (2011)* developed cpSSRs that were subsequently used together with cpSNP markers (developed by *Kane et al., 2012*) to untangle the genetic origins of the Trinitario cultivar in Trinidad and Tobago (*Yang et al., 2013*).

The most commonly used cpDNA intergenic spacer is *trnH-psbA,* which has shown high variability and can be used to elucidate genetic relationships at the intraspecific level (*Azuma et al., 2001*; *Hamilton, Braverman & Soria-Hernanz, 2003*). The *trnH-psbA* region sequences from 10 cacao accessions deposited in the NCBI database produced only one haplotype (*Kane et al., 2012*), whereas *Jansen et al. (2011)* reported the presence of polymorphic sites, which produced a different haplotype. The main polymorphisms reported in the noncoding cpDNA region are inversions, transitions and transversions (*Whitlock, Hale & Groff, 2010*; *Zeng et al., 2012*). Few studies have reported the presence

of insertions or deletions (indels), although indels are probably a common feature in the *trnH-psbA* spacer (*Aldrich et al., 1988*).

Nonetheless, the use of indels for diversity and phylogenetic analysis has been questioned (*Bieniek, Mizianty & Szklarczyk, 2015*; *Whitlock, Hale & Groff, 2010*) because the mechanism causing indels remains unclear. However, indels are informative characteristics because genetic variability detected using polymorphism due to indels or substitutions can be studied without distinction (*Nei, 1987*). Therefore, indels are useful markers. Moreover, the inclusion of indels in diversity and phylogenetic analyses enhances the discriminant power between species (*Raymúndez et al., 2002*; *Hamilton, Braverman & Soria-Hernanz, 2003*; *Kress & Erickson, 2007*; *Sun et al., 2012*) and even between conspecific individuals (*Pérez-Jiménez et al., 2013*). Therefore, the aim of this study was to evaluate the genetic diversity and describe the relationship between individuals of the *Theobroma cacao* L. Criollo type of the Soconusco region (Chiapas, Mexico) using the variations in chloroplast DNA revealed by the *trnH-psbA* spacer sequence.

## MATERIAL & METHODS

### Plant material and sample collection

A total of 45 cacao samples were included in this study. Thirty-eight trees were sequenced for the *trnH-psbA* spacer and analyzed, and seven sequence accessions were downloaded from GenBank as references. A total of 28 of the 38 sequenced trees were selected from plantations in Soconusco (Chiapas, Mexico) based on a previous characterization (*Vázquez-Ovando et al., 2014*) using 10 SSR molecular markers. The individuals were selected based on fruit (pod) and seed traits that resembled those of the Criollo variety. The pods were elongated, deeply grooved and pointed at the apical end and had a lumpy surface with a warty exterior appearance. The seeds had white or slightly pigmented cotyledons that were enveloped in sweet pulp. In agreement with the report by *Vázquez-Ovando et al. (2014)*, the individuals were classified as 12 trees with high Criollo ancestry, 11 Non-Criollo group trees and five admixtures (Table 1). Additionally, 10 accessions were sequenced and included as references: two Forastero variety (Catongo and EET 399), one Trinitario variety (RIM 24) and seven wild Criollo (one collected in the Lacandon rainforest (SL01) and six obtained from the germplasm of the Instituto Nacional de Investigaciones Forestales, Agrícolas y Pecuarias, México (Yaxcabá, Xocen, Lacandón 06, Lacandón 28, Lagarto and Carmelo); Table 1). *Theobroma bicolor* was used as the outgroup. Leaves were collected from trees aged approximately 30 years and placed in plastic bags, taken to the laboratory (4 °C) and stored at −20 °C prior to processing.

### DNA extraction, amplification and sequencing

The total DNA extraction was performed by modifying the method described by *Doyle & Doyle (1990)*. The leaves were washed with sterile water and 70% ethyl alcohol. Approximately 200 mg of cacao leaves were ground with liquid nitrogen with 60 mg polyvinyl pyrrolidone and 1 mL of CTAB buffer (2% CTAB (*w/v*), 20 mM ethylenediaminetetraacetic acid (EDTA), 1.4 M NaCl, 100 mM Trizma® base, pH adjusted to 8 with HCl and 1% 2-mercaptoethanol (*v/v*)). DNA extractions were performed with

**Table 1 Geographic populations\* and genetic classification of the analyzed *Theobroma cacao* trees.** For populations 1–5 (from farms in Soconusco, Mexico) genetic clustering was based on membership to the Criollo group (%) described by *Vázquez-Ovando et al. (2014)* using SSR markers. For the reference trees\*\* (populations 6–9), the genetic grouping was suggested by *Avendaño-Arrazate et al. (2010)* and the database accessions (ICGD; *Turnbull & Hadley, 2016*).

| Pop\* | Coordinates latitude (N)/longitude (W) | Criollo ($n = 20$) | Non-Criollo ($n = 16$) | Admixtures ($n = 9$) |
|---|---|---|---|---|
| 1 | 14°59′28″N, 92°26′44″W (Huehuetán) 14°52′55″N, 92°21′42″W (Tapachula) | TASG12 (93%) TASG18 (95%) | HUJF01 (9%) HUJF03 (2%) | TASG16 (86%) |
| 2 | 14°56′41″N, 92°09′59″W (Tuxtla Chico) 14°59′53″N, 92°10′44″W (Cacahotán) | TCHR04 (98%) | CAAM12 (1%) | CAAM04 (53%) |
| 3 | 14°47′31″N, 92°11′11″W (Frontera Hidalgo) 14°38′27″N, 92°13′47″W (Suchiate) | | FHSA06 (1%) SUED02 (2%) SUED03 (1%) SUED06 (1%) | FHSA02 (36%) |
| 4 | 14°48′56″N, 92°29′06″W (Mazatán) | MAMG12 (98%) | MAMG03 (2%) MAMG04 (1%) MAMG07 (1%) MAMG08 (9%) | MAMG10 (24%) |
| 5 | 15°28′07″N, 92°48′42″W (Mapastepec) 15°10′31″N, 92°38′06″W (Villa Comaltitlán) 15°11′17″N, 92°36′55″W (Villa Comaltitlán) | MAJH02 (96%) VCHL01 (97%) VCHL02 (96%) VCHL03 (97%) VCHL04 (97%) VCLB02 (97%) VCLB03 (98%) VCLB04 (98%) | | MAJH03 (63%) |
| 6\*\* | 20°32′29.25″N, 88°50′35.82″W (Yucatán) | Yaxcabá Xocen | | |
| 7\*\* | 16°06′42.92″N, 90°56′31.28″W (Selva Lacandona) | Lacandón 06 Lacandón 28 SL01 | | |
| 8\*\* | INIFAP (Several) | Lagarto Carmelo | CATONGO EET 399 | RIM 24 |
| 9\*\* | Accessions (ICGD) | CRIOLLO 22 | SCA 6 (MIA 29885) AMELONADO (TARS 16542) MATINA 1/6 | ICS 1 (TARS 16656) ICS 6 (TARS 16658) ICS 39 (TARS 16664) |

Notes.
ICGD, International Cocoa Germplasm Database; TARS, Tropical Agriculture Research Station; INIFAP, Instituto Nacional de Investigaciones Forestales, Agrícolas y Pecuarias.

chloroform-isoamyl alcohol and precipitated with isopropanol. The extracted DNA was then purified with a mixture of phenol:chloroform:isoamyl alcohol (25:24:1). The DNA was dissolved in 60 μL of Milli-Q water and its integrity verified on 0.8% agarose gels. The purity was obtained from the 260/280 absorbance ratios while quantifications were

estimated from absorbances at 260 nm. Absorbance readings were performed on Jenway, Genova Spectrophotometer (Krackeler Scientific Incorporation, Albany, NY, USA).

The cpDNA amplification of the *trnH-psbA* intergenic spacer was conducted using the forward primer 5′-CGCGCATGGTGGATTCACAATCC-3′ and reverse primer 5′-GTTATGCATGAACGTAATGCTC-3′ (*Shaw & Small, 2005*). The PCR conditions were modified from *Shaw & Small (2005)*. The PCR was performed in a 25 μL reaction mixture containing 100 ng of genomic DNA, 4 μL of 10x PCR ViBuffer A (Vivantis$^{TM}$ Oceanside CA, USA), 1 μL of MgCl$_2$ (50 mM), 0.5 μL of dNTP Mix (10 mM, Promega), 0.05 mM of each primer and 2.5 U of Taq DNA polymerase (Vivantis$^{TM}$). Following one cycle of 5 min at 94 °C, 35 PCR cycles of 30 s at 94 °C, 30 s at 53 °C and 1 min at 72 °C and a 10 min 72 °C final extension were performed in a TC3000 thermal cycler (Techne, Cambridge, UK). To verify the presence of amplicons, the PCR products were separated on 6% polyacrylamide gels using 0.5X TBE buffer at 110 V for 210 min, stained with ethidium bromide (0.6 ng/μL) for 15 min, visualized under UV light and photographed with a Gel Doc$^{TM}$ EZ Imager gel documentation system (Bio-Rad, USA). Fragment sizes were estimated using Image Lab (v. 4.0.1, Bio-Rad Laboratories) and integrating the GeneRuler$^{TM}$ 100 bp DNA Ladder Plus (Fermentas$^{®}$) as a molecular weight marker.

The PCR products were directly sequenced using the Dye Terminator Cycle Sequencing with Quick Start Kit (GenomeLab$^{TM}$) on a CEQ$^{TM}$ 8000 automatic DNA sequencer (Beckman Coulter$^{TM}$). To validate the results, the DNA was extracted twice and amplified independently. The sequences were verified by comparison with their forward and reverse sequences when applicable.

## Sequence alignment and data analysis

The sequence quality was checked and the electropherograms were edited using BioEdit© (*Hall, 1999*). Sequences were limited at the ends to avoid the presence of variable sites due to the introduction of sequencing artifacts by the polymerase (approx. 40 bp) and aligned with ClustalW 1.81 (*Thompson, Higgins & Gibson, 1994*). Visual inspection and manual editing of the sequences was performed to confirm the variable sites. We used two different analytical approaches based on the geographic origin and the genetic origin of the samples (Table 1). In both approaches, molecular diversity indices including the number of segregating sites ($S$), the number of haplotypes, the haplotype diversity ($Hd$) and the nucleotide diversity ($\pi d$) were estimated following the methods of *Nei (1987)* in DnaSP© 5.1 (*Rozas et al., 2010*).

To infer evolutionary relationships at the intraspecific level, we produced a network. The method used was median-joining (MD) based on parsimony criteria (*Bandelt, Forster & Röhl, 1999*; *Polzin & Daneshmand, 2003*) and was performed with the software Network© 4.6.1.3 (*Bandelt et al., 1995*).

Analysis of molecular variance (AMOVA), pairwise *Fst* values and statistical analyses of molecular variance ($F_{CT}$ (test performed by permuting individuals within populations), $F_{ST}$ (test performed by permuting genotypes among populations but within groups) and $F_{SC}$ (test performed by permuting genotypes among groups)) were estimated using Arlequin© version 3.0 (*Excoffier, Laval & Schneider, 2005*). Significance was evaluated

**Table 2** Nucleotide polymorphic sites and cpDNA haplotypes in cacao populations based on variation in the intergenic *trnH-psbA* spacer region.

| Haplotype | Polymorphic site | | | | | | Populations (see Table 1 note) | | | | | | | | |
|---|---|---|---|---|---|---|---|---|---|---|---|---|---|---|---|
| | 22 | 134 | 206 | 309 | 310 | 487 | Pop1 | Pop2 | Pop3 | Pop4 | Pop5 | Pop6 | Pop7 | Pop8 | Pop9 |
| H1 | – | T | – | A | A | A | 3 | 2 | 3 | 2 | 5 | 1 | 3 | | |
| H2 | C | T | – | A | A | A | 1 | | 1 | | 2 | | | 1 | |
| H3 | C | T | A | – | – | – | | | 1 | | 1 | | | | |
| H4 | – | A | – | A | A | A | | | | 1 | | | | | |
| H5 | – | T | – | A | A | – | | | | | | 1 | | | |
| H6 | C | T | – | A | – | A | 1 | | | | | | | | |
| H7 | – | T | – | A | – | – | | | | 1 | | | | 1 | |
| H8 | – | T | A | – | – | A | | | | 1 | | | | 1 | |
| H9 | – | T | A | – | – | – | | | | 1 | 1 | | | 1 | |
| H10 | – | A | – | A | A | – | | 1 | | | | | | | |
| H11 | – | T | – | A | – | A | | | | | | | | 1 | 1 |
| H12 | – | T | A | A | – | – | | | | | | | | | 6 |

by 99,999 random sequence permutations. To determine whether sample sites clustered on a population level, a spatial analysis of variance (SAMOVA) was conducted (*Dupanloup, Schneider & Excoffier, 2002*) using haplotype data and the geographic coordinates of each of the 5 sample sites. The SAMOVA was run for $K = 2$–5 putative populations to determine the maximum $F_{ST}$ value and the highest proportion of differences between populations due to genetic variation.

The neutral evolution of chloroplast DNA was evaluated to examine whether any population had experienced historic demographic changes using Tajima's $D$ test (*Tajima, 1989*) with Arlequin© version 3.0 (*Excoffier, Laval & Schneider, 2005*). It was computed for all seven geographic populations and overall without *a priori* populations designation; $p$-value were generated using 1,000 simulations under a model of selective neutrality.

Seven accessions from the NCBI database were included as references in the genetic origin approach analysis: MATINA 1/6 (HQ336404.2), CRIOLLO-22 (JQ228379.1) AMELONADO (JQ228380.1) SCA 6 (JQ228382.1), ICS 1 (JQ228381.1), ICS 6 (JQ228383.1) and ICS 39 (JQ228387.1).

## RESULTS

### Sequence characterization and genetic diversity

The *trnH-psbA* intergenic spacer sequences from 45 *Theobroma cacao* samples (Table 1) were aligned with a consensus length of 526 bp. Six segregating polymorphic sites (Table 2) were present as five indels (Fig. 1) and one transversion (T ↔ A event at position 134). These polymorphisms resulted in 12 haplotypes over all samples, of which four were singletons represented by a unique sequence in the sample (Table 2). The nucleotide composition of the fragment revealed that it was AT-rich (A + T, 75.52%). The sequences determined in this study were deposited in GenBank under accession numbers KU061021–KU061059.

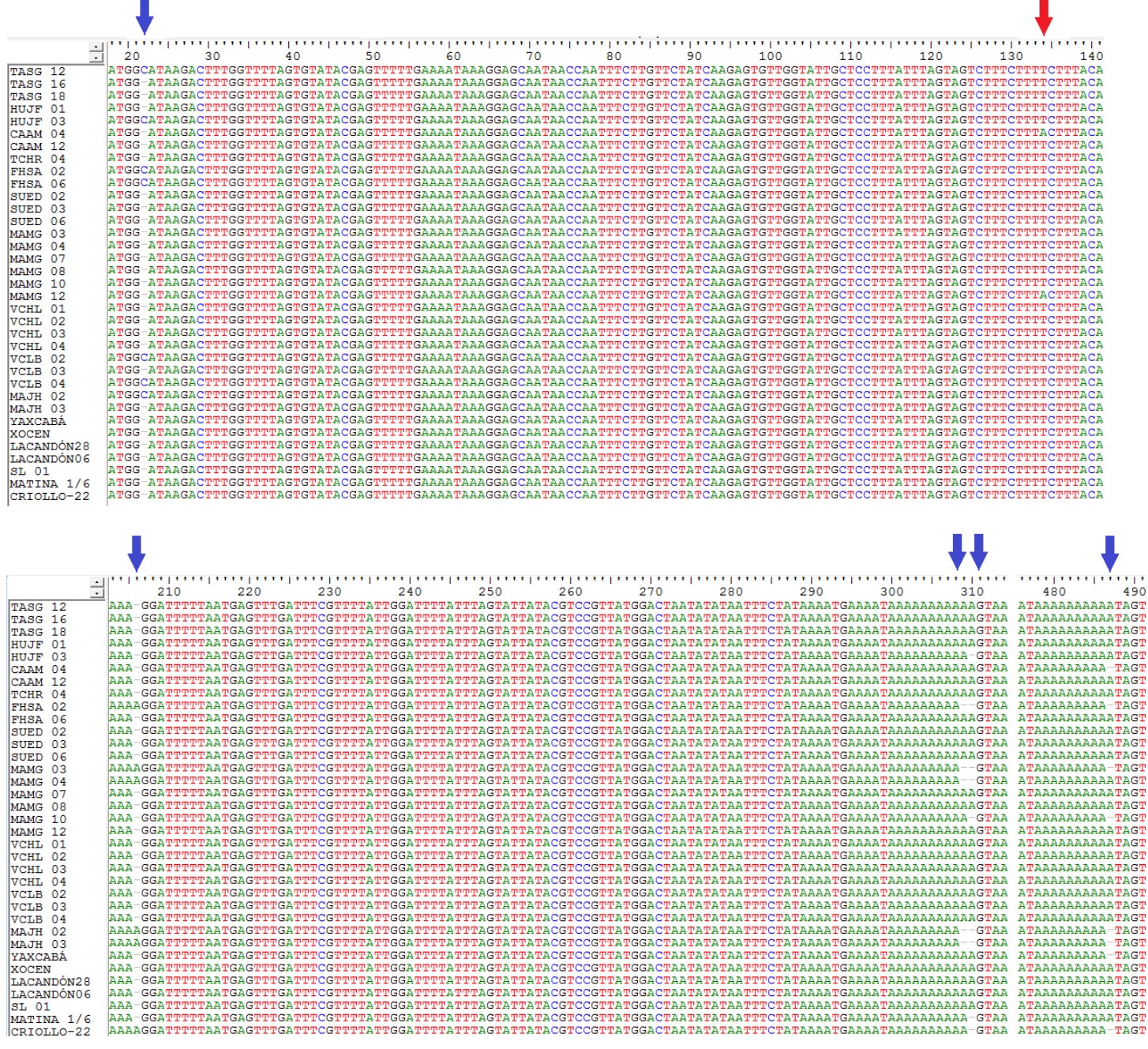

**Figure 1   Location of indels (blue arrows) and the transversion (red arrow) in a sequenced fragment of the chloroplast DNA *trnH-psbA* intergenic spacer from *Theobroma cacao* trees.** See Table 1 for sample details.

The results based on the geographic approach revealed that the overall average haplotype diversity ($Hd$) and nucleotide diversity ($\pi d$) values were 0.66 and 0.0032, respectively (Table 3). Under this approach 10 haplotypes were identified. The most frequent haplotype (H1) was shared by 19 trees from seven geographic populations formed *a priori* (Table 2). Four trees belonging to Population 1 (one tree), Population 3 (one tree) and Population

**Table 3** Genetic diversity in cacaos from Soconusco (Chiapas, Mexico) grouped by the geographic approach (Pop) and genetic origin approach.

| Pop | Locality | N | S | Sn | H | Hd | πd |
|---|---|---|---|---|---|---|---|
| 1 | Huehuetán, Tapachula | 5 | 2 | 1 | 3 | $0.70 \pm 0.21$ | $0.0019 \pm 0.0017$ |
| 2 | Cacahoatán, Tuxtla Chico | 3 | 2 | 1 | 2 | $0.67 \pm 0.31$ | $0.0026 \pm 0.0026$ |
| 3 | Frontera Hidalgo, Suchiate | 5 | 5 | 0 | 3 | $0.70 \pm 0.21$ | $0.0042 \pm 0.0032$ |
| 4 | Mazatán | 6 | 5 | 3 | 5 | $0.93 \pm 0.12$ | $0.0048 \pm 0.0035$ |
| 5 | Mapastepec, Villa Comaltitlán | 9 | 5 | 0 | 4 | $0.69 \pm 0.14$ | $0.0039 \pm 0.0027$ |
| 6 | Yucatán | 2 | 1 | 1 | 2 | $1.00 \pm 0.50$ | $0.0019 \pm 0.0027$ |
| 7 | Selva Lacandona | 3 | 0 | 0 | 1 | 0 | 0 |
| Total | | 33 | – | 6 | – | | |
| Mean ± sd | | | | | | $0.66 \pm 0.08$ | $0.0032 \pm 0.0021$ |
| Genetic origin approach[*] | | | | | | | |
| "Criollo" | | 12 | 6 | 1 | 4 | $0.64 \pm 0.13$ | $0.0025 \pm 0.0019$ |
| "Non-Criollo" | | 11 | 5 | 1 | 5 | $0.62 \pm 0.16$ | $0.0030 \pm 0.0021$ |
| "Admixtures" | | 5 | 5 | 1 | 5 | $1.00 \pm 0.12$ | $0.0060 \pm 0.0041$ |
| Criollo-reference[a] | | 8 | 4 | 1 | 5 | $0.79 \pm 0.15$ | $0.0033 \pm 0.0025$ |
| Forastero-reference[a] | | 5 | 3 | 0 | 3 | $0.80 \pm 0.16$ | $0.0031 \pm 0.0025$ |
| Trinitario-reference[a] | | 4 | 4 | 0 | 2 | $0.50 \pm 0.27$ | $0.0038 \pm 0.0032$ |
| Total | | 45 | – | 4 | – | | |
| Mean ± sd | | | | | | $0.80 \pm 0.05$ | $0.0038 \pm 0.0024$ |

**Notes.**

$N$, Samples sizes; $S$, Number of segregating; $Sn$, Singletons; $H$, Number of haplotypes; $Hd$, Haplotype diversity; $\pi d$, Nucleotide diversity; $sd$, standard deviation.

[a] Including sequences GenBank (Criollo-reference $n = 1$, Forastero-reference $n = 3$, Trinitario-reference $n = 3$).

[*] Classification based on membership (>90%) to Criollo type, see Table 1 (*Vázquez-Ovando et al., 2014*).

5 (two trees) formed the second most common haplotype (H2). Overall, 60% of the haplotypes (six of the ten) were singletons (Fig. 1). The analysis showed that most of the genetic diversity was found in Population 4 (Mazatán), which contained the highest values for the most informative indices (Table 3); Population 4 included 50% of the identified haplotypes (Fig. 1). The other populations maintained moderate Hd and low $\pi d$ values that were similar for each population (Table 3). The Yucatán and Selva Lacandona populations (wild) exhibited Hd values of 1 and 0, respectively, although these data like those of Pop 2, would be influenced by the low numbers of reference individuals.

When the data analysis was based on the genetic origins, the highest Hd (1.0) was found in the Admixture group (Table 3). In contrast, the Trinitario-reference group had the lowest *Hd* value (0.5). The $\pi d$ was low (0.0025–0.006) for all groups, which was similar to the results obtained with geographic approach. The Forastero-reference and Trinitario-reference groups did not present singletons (Table 3). Sequences from the NCBI database were grouped into one haplotype (H12) with the exception of MATINA 1/6, which grouped in H11 with EET 399 corresponding to the Forastero-reference group.
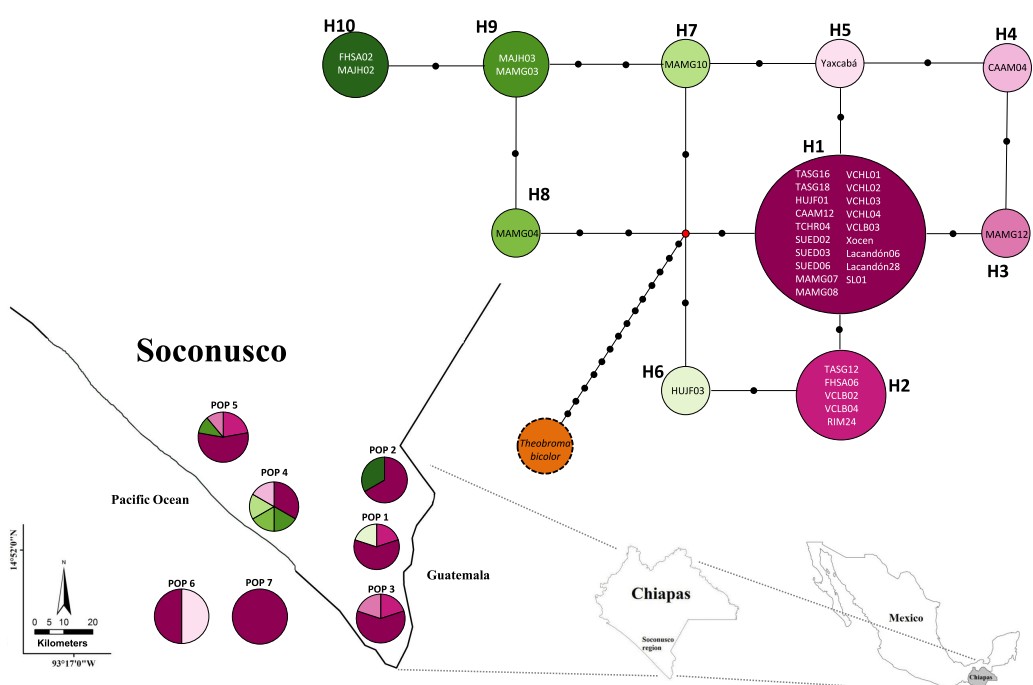

**Figure 2** Median joining network for chloroplast DNA *trnH-psbA* intergenic spacer haplotypes of *Theobroma cacao* trees from Soconusco, Mexico, and the outgroup haplotype (*Theobroma bicolor*). The map indicates the geographic distribution of the haplotypes. The colored portions represent the proportions of the same haplotype occurring in each sampling locality. Trees employed as the references (Pop 6 and Pop 7) are shown outside the map. The population code and details are shown in Table 1.

## Intraspecific relationships

Figures 2 and 3 show the haplotype networks built with data from the geographic (Fig. 2) and genetic approaches (Fig. 3). The individuals belonging to each haplotype are also included. The general base has a common haplotype for the two networks (H1) that includes cacao trees from all geographic populations (Fig. 2) and four of six groups based on the genetic approach (Fig. 3). A unique set of intermediate haplotypes are derived from this common haplotype (H1) and are interconnected by one or two mutational steps in both networks. The H4–H6 haplotypes were farthest from the central clade (i.e., newly created haplotypes; Figs. 2 and 3). Haplotypes H3–H6 were singletons.

## Population genetic structure

The analysis of molecular variance (AMOVA) was not significant and had a value of $F_{ST} = 0$. In the spatial analysis of molecular variance (SAMOVA), the value $K = 2$ extended the $F_{ST}$ to 0.0439 and generated two groups: the first contained only Population 4 (Mazatán) and the second grouped the other geographic populations (Table 4).

The neutrality tests showed non-significant values in the Tajima's $D$ for all populations. Although, in Population 4 the Tajima's $D$ value was negative ($D = -0.93302$) because this population including a transversion. All other populations showed values of $D = 0$; however, the overall value for this test was $D = -0.13329$ ($P > 0.1$).

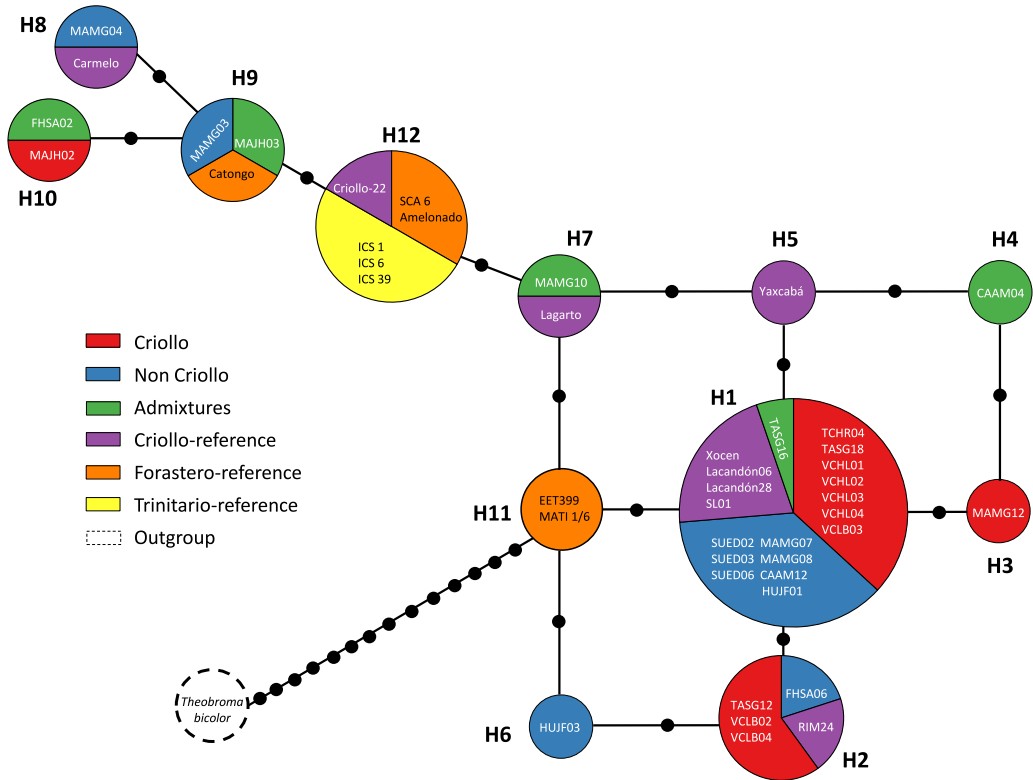

**Figure 3** Median joining network for the chloroplast DNA *trnH-psbA* intergenic spacer haplotypes of *Theobroma cacao* trees cultivated in Soconusco, Mexico, and the reference accessions. The circle sizes are proportional to the haplotype frequencies, and the color represents the proportions of the same haplotype occurring in each genetic group. For genetic group details, see Table 1.

**Table 4** Spatial analysis of molecular variance ($K = 2$) for cacao populations and the statistical analysis of molecular variance fixation indices corresponding to the groups.

| Source of variation | df | SS | VC | Variation (%) | Fixation indices | P value |
|---|---|---|---|---|---|---|
| Among groups | 1 | 1.61 | 0.1282 | 13.98 | $F_{SC} = -0.1115$ | 0.7341 |
| Among populations within groups | 5 | 2.51 | −0.0879 | −9.59 | $F_{ST} = 0.0439$ | 0.0068 |
| Within populations | 26 | 22.80 | 0.8765 | 95.61 | $F_{CT} = 0.1398$ | 0.1496 |
| Total | 32 | 26.91 | 0.9168 | | | |

**Notes.**
df, degrees of freedom; SS, Sum of squares; VC, Variance components.

## DISCUSSION

In this study, high haplotype variation was found in the chloroplast DNA from cacao trees grown in the Soconusco region. No inversions or transitions were found, although they were reported to be common in other plants (*Whitlock, Hale & Groff, 2010*; *Zeng et al., 2012*). However, we found five insertions or deletions (indels) in three poly-A regions and one A ↔ T transversion (Fig. 1). This result agreed with the findings reported by *Jansen et al. (2011)* in the MATINA 1/6 accession and supported the affirmation by
*Aldrich et al. (1988)* that indels were a presumably common feature in the *trnH-psbA* region. In the data analysis, we included the indels as informative character states, and the high interspecific divergence of the spacer region allowed their use as a marker for DNA barcoding (*Kress & Erickson, 2007*). The molecular diversity indices determined in the present study were similar to the results of *Zeng et al. (2012)* using the same intergenic spacer, which revealed 11 haplotypes for 35 *Thinopyrum intermedium* samples, low nucleotide diversity ($\pi d = 0.00473$) and moderately high haplotype diversity ($Hd = 0.7331$) (our results for the geographic populations were $\pi d = 0.0032$ and $Hd = 0.66$). The results of those authors supported the use of one intergenic spacer to reveal nucleotide polymorphisms similar to our findings.

Our haplotype diversity results are contrary to those reported by *Vázquez-Ovando et al. (2014)*. These authors reported low genetic diversity in individuals from the same region (in particular Population 4 in Mazatán) using microsatellite markers. One reason for the discrepancy may be that a larger number of individuals with Criollo ancestry was included in that study, resulting in a higher degree of homozygotes and lower population genetic diversity. Our study also included individuals from other cacao varieties that possessed greater genetic diversity, at least at the nuclear DNA level. However, the low nucleotide diversity found in this study was supported by the low genetic variability found using nuclear microsatellites. Individuals included in both studies showed great morphological pod variability that resembled the Criollo type (e.g., different degrees of roughness, color and deep grooves) reported previously by *Avendaño-Arrazate et al. (2010)*. This finding could reveal a greater association between the morphological variability of the cacao pod with the reported allelic richness (*Vázquez-Ovando et al., 2014*) and the presence of polymorphic sites in several trees found in our study.

The number of haplotypes was higher than the number of polymorphic sites (Table 2). This feature is associated with ancestral species that have sufficiently diverged to accumulate mutations among different haplotypes (*Roger, 1995*). The haplotype number detected in the present study is unusually striking compared with other works. For example, *Yang et al. (2013)* found only three haplotypes based on three cpSNP markers. However, that study exclusively analyzed nucleotide substitutions, whereas in this study five indel regions were included; this difference may explain the high haplotype diversity found here. Indels have been reported to have a high mutation rate compared with other regions of the cpDNA (*Ingvarsson, Ribstein & Taylor, 2003*), especially when they are repeated locally (*Yamane, Yano & Kawahara, 2006*) such as in region 309–310 of our sequences (Fig. 1).

Several explanations are possible for the presence of more than one Criollo haplotype. First, only the maternal line gave rise to the eight Criollo haplotypes by mutation. Second, the "Criollo" phenotype had multiple provenances, indicating that the ancient haplotypes persisted over time in the Soconusco cacao farms. Third, some samples were misclassified as "Criollo" (especially MAJH02 and Carmelo, which were the most divergent "Criollo" individuals; haplotypes 4 and 6, respectively, Fig. 3). These samples possibly belong to the Admixture group rather than the Criollo. However, they are also contenders for the Modern Criollo group (i.e., individuals classified as Criollo that might have been introgressed with Forastero genes) (*Motamayor et al., 2002*) and preserve phenotypic traits of the ancient

Criollo. Finally, heteroplasmy and haplotype polymorphisms of plastid genomes within and among individuals were documented in Malvaceae (*Wolfe & Randle, 2004*). These phenomena could be present in *Theobroma cacao*. To test those hypotheses, additional studies are needed using high-throughput sequencing of chloroplast genomes.

Population 7 (Selva Lacandona) exhibited no haplotype diversity ($Hd = 0$). However, haplotype H1 located in this population is considered the common ancestor because it is shared by all populations (Fig. 2). In contrast, the two individuals belonging to Population 6 (Yucatán), which exhibited different haplotypes (H1 and H5) from one another, were interrelated by only a mutational step (Fig. 2). This result shows that an individual tree belonging to a Yucatán population (as well as all other haplotypes) eventually descended from other individuals in this region where the Maya people grew cacao.

The low nucleotide polymorphism levels could be explained by rapid population expansion events in the distribution range, whereas high haplotype diversity might be due to the continuous introduction of individuals from different locations. However, these results should be interpreted with caution because, as stated above, they are limited by the sample's size. Populations recently introduced or expanded from a small number of founders would have a common haplotype shared by most individuals and many rare haplotypes connected to the main population by a few independent mutations (*Slatkin & Hudson, 1991*; *Avise, 2000*) such as observed in the present study (Fig. 2). A similar argument was proposed based on the use of microsatellite markers (*Vázquez-Ovando et al., 2014*).

The relatively low variability in the cultivated cacao populations was supported by the lack of neutrality revealed by the global Tajima test. Specifically, the negative Tajima's *D* value ($-0.93302$) in Population 4 (Mazatán) could be related to a "bottleneck" event, which would indicate population expansion and not natural expansion because it was a cultivated population. The occurrence of unclear events in the past (disease, volcanic eruptions or other natural events) may have caused the almost complete disappearance of populations established by the people in the Mesoamerican region (*De Sahagún, 2009*, *Codex Florentino*). Rapid expansion due to recolonization of the populations and the probable introduction of other varieties of cacao trees not native to the region would have subjected the populations to a bottleneck events in very recent periods. However, these are presumptive weak inferences of the population history based on a single locus. The bottleneck event could also be related to the loss of alleles (haplotypes; especially rare alleles), which is much greater than the loss of genetic variance *per se*. Although these rare alleles contribute little to the total genetic variability, they can provide unique responses against evolutionary challenges similar to the high number of unique haplotypes found in this study (3 singletons in Population 4). The presence of both common and rare haplotypes can be the result of a directional-purifying selection process or expansion events from small populations (*Hedrick, 2005*). The H3–H8 haplotypes (cultivated populations) are singletons. This finding agreed with *Crandall & Templeton (1993)*, who reported that the singletons tend to are connected to haplotypes from the same population. Population 4 (Mazatán) shows the highest haplotype diversity, which makes this population an important reservoir of genetic material at the chloroplast and possibly phenotypic levels based on the abundance of pod morphologies observed in this population.

Overall, cacao trees with high Criollo ancestry were located in the center of the haplotype network. This result was supported by the coalescence theory that predicted that the ancient haplotype should be the most common and most distributed among the populations. In concordance, derived haplotypes would be less frequent and in many cases would be private; these haplotypes would be located in regions containing the latest cultivated cacao populations. The H10 and H9 haplotypes may have been recently created because they are located at the ends of the network, possibly due to germplasm exchange with traits of interest to cacao farmers. These anthropogenic activities may have had a strong impact on the levels of variation observed in the cpDNA sequences, which explains the observed lack of differentiation. Additionally, migration over long distances and the exchange by farmers contributed to the colonization of new regions founded by a few individuals, thereby establishing different alleles via mutation and genetic drift.

Furthermore, the $F_{ST} = 0$ value determined by AMOVA revealed that all of the molecular variance occurred within populations. Indeed, the SAMOVA $F_{ST}$ value (Table 4) was not sufficient to show at least moderate differentiation between populations ($F_{ST} \geq 0.05$). This finding provides some explanations regarding the demographic history of *T. cacao* trees, indicating that the populations formed *a priori* and experienced gene flow, resulting in population homogenization. The spatial analysis revealed the highest differentiation between groups when $K = 2$ was tested; $K = 3$ ($F_{ST} = 0.00088$) grouped trees from the Yucatán, Selva and Cacahoatán in the same genetic population. This grouping is unusual because the geographic distance is longer among the three localities and may be associated with the distribution of trees in the past (i.e., the ancestral haplotype (H1) grouped individuals from Selva; one mutational step resulted in the origination of the individuals from the Yucatán, which in turn originated the individuals at Cacahoatán by the same event (Fig. 3)). Following this criterion, H4 and the non-Criollo trees belonging to H1 have a greater correspondence with the Criollo genotype, although it was previously reported to be an Admixture and non-Criollo, respectively (*Vázquez-Ovando et al., 2014*).

## CONCLUSIONS

Indels and one transversion located in the chloroplast DNA *trnH-psbA* spacer region of cacao trees can distinguish individuals that are indistinguishable in other marker systems or separated by only few SSR markers, and further support use of these cpDNA markers. The molecular analysis showed low nucleotide diversity but high haplotype diversity possibly due to population bottleneck events. These results were confirmed by the negative Tajima's *D* and the arrangement of the haplotype network. We identified 10 different haplotypes (from cultivated trees) of which H3–H8 resulted in singletons because they were not associated with other cacaos or with those reported in the molecular databases. The presence of these haplotypes accompanied by the low number of mutational steps might suggest a very short evolutionary history or events that led to disappearing-expanding populations in southern Mexico. These results suggest confirmation of selection of fruits from few mother trees (even as few as one) that were moved by human agents from South America into Mexico and that the Criollo complex may be homogenous based on maternal

influence. Increasing sampling numbers would go a long way in establishing whether a true sub-structure of maternal origin exists. One geographic population (Pop 4, Mazatán) consisted of high frequency haplotypes, which makes this zone an important reservoir of genetic material at the chloroplast and possibly phenotypic levels because an abundance of pod morphology was also observed in this population. The genetic differentiation between populations was zero, suggesting that gene flow homogenized the populations.

## ACKNOWLEDGEMENTS

The authors thank Nancy Gálvez-Reyes for her advice on data analysis and comments on the manuscript. The authors also thank the three referees for their exhaustive revisions, which helped to improve the manuscript.

### Funding
This work was partly funded by SEP-Mexico through the program PROFOCIE-2014-07MSU0001H-11 and by Consejo Estatal de Ciencia y Tecnología del Estado de Chiapas, Mexico. The funders had no role in study design, data collection and analysis, decision to publish, or preparation of the manuscript.

### Grant Disclosures
The following grant information was disclosed by the authors:
SEP-Mexico: PROFOCIE-2014-07MSU0001H-11.
Consejo Estatal de Ciencia y Tecnología del Estado de Chiapas, Mexico.

### Competing Interests
The authors declare there are no competing interests

### Author Contributions
- Nidia Gutiérrez-López performed the experiments, analyzed the data, wrote the paper, prepared figures and/or tables.
- Isidro Ovando-Medina conceived and designed the experiments, contributed reagents/materials/analysis tools, wrote the paper, reviewed drafts of the paper.
- Miguel Salvador-Figueroa conceived and designed the experiments, contributed reagents/materials/analysis tools, reviewed drafts of the paper.
- Francisco Molina-Freaner and Carlos H. Avendaño-Arrazate contributed reagents/materials/analysis tools, reviewed drafts of the paper.
- Alfredo Vázquez-Ovando conceived and designed the experiments, analyzed the data, contributed reagents/materials/analysis tools, wrote the paper, prepared figures and/or tables, reviewed drafts of the paper.

### DNA Deposition
The following information was supplied regarding the deposition of DNA sequences:
GenBank accession numbers KU061021–KU061059.

## Data Availability

The research in this article did not generate any raw data.

## Supplemental Information

Supplemental information for this article can be found online at http://dx.doi.org/10.7717/peerj.1855#supplemental-information.

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
