# Peer review of "Unique haplotypes of cacao trees as revealed by trnH-psbA chloroplast DNA"

_PeerJ, doi:10.7717/peerj.1855_

## Round 0.1 · original submission · Major Revisions

I agree with Reviewer 3 that parts of the manuscript are difficult to understand due to the quality of the English. I suggest you have the manuscript edited by someone who can help with the English.

Major revisions are required to clarify the presentation and interpretation of the results, particularly to address the criticisms of Reviewer 3. Carefully consider the comments of all three reviewers in your revision, including those in the annotated manuscript submitted by Reviewer 1.

In addition, I question your statement that no transitions or tranversions were observed. A<->T is a transversion.

·

Basic reporting

The study of Gutiérrez-López et al. evaluate the genetic diversity and describe the relationship of cacao from the Soconuscoregion of Mexico, using chloroplast DNA. The article is in general well written and provide conclusive results.
The introduction gives good background informationof the importance of the work.
The authors refer to the classification of cacao. One based on morphological traits and geographical origin (morphogeographical):Criollo, Forastero and Trinitario. The other based on SSR with 10 genetic groups. But in the article indicate that the genetic origin of the plants is Criollo, Forastero and Hybrid, and in also use the classification Criollo, Non Criollo and admixtures. It is more accuracy and consistent with the Introduction indicate as morphogeographical origin (or groups) Criollo, Forastero and Trinitario in all the article.
Figures are relevant to the content of the article. In the case of Figure 1 I think that could be more illustrative incorporate the sequence of all plants from polymorphic site 22 to 487, not only from 189 to 487. In Figure 2 the authors could indicate in the map the geographical localization of Pop6 and Pop7.

Experimental design

The research is relevant and the experimental design is appropriate. The description of materials and methods used include enough information to be reproduced.

Validity of the findings

The article does not indicate the repository were the data are available.
The conclusions are appropriated and related to the objective.

Reviewer 2 ·

Basic reporting

This manuscript reported the unique haplotypes in 28 cacao trees from different farms in the Soconusco region of Mexico. It was investigated by sequence variation of chloroplast DNA trnH-psbA intergenic spacer. According to this research, it is clear that six polymorphic sites were detected, including five insertion/deletion (indels) and one substitution. The indels discovered in the intergenic spacer trnH-psbA of cacao trees should be a worthy scientific contribution to genetic markers development and Mazatán population displaying high frequency of haplotypes could be an important resource for genetic improvement of cacao trees. In general, these findings are important to document, but the English written should be improved and there are a number of ways that this manuscript could be stronger.

Following are some corrections to English grammars errors and typos that will help improve this manuscript:

L55-56: “Based on simple sequence repeat (SSR) analysis, Motamayor et al. (2008) propose ten genetic groups.” should be “Based on simple sequence repeat (SSR) analysis, Motamayor et al. (2008) propose ten genetic groups of Criollo cacao.”
L60-61:”… to low in natural populations (Whitkus et al., 1998; by using RAPD markers), and cultivated forms (Vazquez-Ovando et al., 2014 , by using microsatellite markers),” should be ”… to low in natural populations (Whitkus et al., 1998; by RAPD markers), and cultivated forms (Vazquez-Ovando et al., 2014, by microsatellite markers),”
L69: “The chloroplast DNA (cpDNA) and markers based on it, they are increasingly used for studies of…” should be “The chloroplast DNA (cpDNA) and markers based on it are increasingly used for studies of…”
L87-89: “Nonetheless, the use of indels for diversity and phylogenetic analysis has been questioned by some authors (Bieniek, Mizianty & Szklarczyk, 2015; Whitlock, Hale & Groff, 2010), because the mechanism by indels are generated remains unclear.” should be “Nonetheless, the use of indels for diversity and phylogenetic analysis has been questioned (Bieniek, Mizianty & Szklarczyk, 2015; Whitlock, Hale & Groff, 2010), because the mechanism causing indels remains unclear.”
L115: “…México [Yaxcabá, Xocen, Lacandón 06 Lacandón 28, Lagarto and Carmelo])” should be “México [Yaxcabá, Xocen, Lacandón 06, Lacandón 28, Lagarto and Carmelo])”
L190: “The geographic approach analysis revealed overall the average values of haplotype diversity…” should be “The results based on geographic approach revealed overall the average values of haplotype diversity…..”
L198-200: “Yucatán and Selva Lacandona populations (wild) they exhibited Hd 1 and 0 respectively, although these data are influenced by the low number reference individuals.” should be “Yucatán and Selva Lacandona populations (wild) exhibited Hd 1 and 0 respectively, although these data were influenced by the low number reference individuals”
L206: “…it grouped in H11 whit EET399, that corresponding to Forastero-reference group.” should be “… grouped in H11 with EET399, that corresponding to Forastero-reference group.”
L226-227: “In this study was found high haplotype variation in chloroplast DNA cacao trees grown in the Soconusco region.” should be “In this study, high haplotype variation in chloroplast DNA was found in cacao trees grown in the Soconusco region.”
L267: “…indicate population expansion, not natural because of it is cultivated populations.” should be “…indicate population expansion, not natural because of it is cultivated population.”
L275-277: “Although these rare alleles contribute little to the total genetic variability, can provide unique responses against challenges evolutionary as found in this study a high number of unique haplotypes” should be “Although these rare alleles contribute little to the total genetic variability, they can provide unique responses against evolutionary challenges, just as a high number of unique haplotypes found in this study.”
L285-287: ”Overall, cacao trees with high ancestry was located in the center of haplotype network, this supported by coalescence theory that predicts the ancient haplotype should be the most common and most distributed among populations.” should be “Overall, cacao trees with high ancestry were located in the center of haplotype network. This result was supported by coalescence theory that predicts the ancient haplotype should be the most common and most distributed among populations.”

Experimental design

One concern is why did the authors select Theobroma bicolor as outgroup? Is it an appropriate outgroup? It should be introduced more to the general readers that the phylogenetic information between T. bicolor and cacao (Theobroma cacao L.).

Another minor concern is the authors said in Discussion (L 240-242): “Our results of the haplotype diversity are contrary, those reported by Vazquez-Ovando et al. (2014) who reported low genetic diversity when the study conducted with individuals of the 242 same region (in particular the Population 4 Mazatán) using microsatellite markers.” But the authors didn’t explain the possible and underlining mechanisms that could result in this sort of contrary. I’m very interested in these possible explanations to this difference.

Validity of the findings

My main concern is about the use on sequence alignment method. I agree with the results based on the Clustal W alignment. However, why did the authors use ClustalW only? Did they have a try to run other alignment methods or software, such as MUSCLE or T-Coffee? Is there any different alignment result among the different alignment methods?

Additional comments

The authors should correct all English typos and grammar errors carefully. No any more comments here.

·

Basic reporting

Intent of manuscript is well founded and is a welcome study in cacao. It represents an area of study that is intriguing particularly as it incorporates one of the early cacao groups that was associated with the early peoples of Latin America. However, the manuscript suffers from poor English usage throughout. Clarity of expression, logical arrangement and sentence structure among others made it difficult at times to unambiguously understand the text including interpretation of the findings. The introduction could be improved by incorporating cacao chloroplast studies that deal with genetic diversity before moving on to trnH-psbA.

Tables should be revisited for clarity of presentation. Reconsider table headings, abbreviations, legends and number of decimal places to improve readability. Figure 2 was difficult to read due to the size of font and the colour choices. The country map was not adequately covered in the legend and there was confusion as to the colour legends for the country map and the haplotype networks that were generated. Subheadings needed to be properly separated from the main text. Referencing was reasonably complete. Please correct Vazquez-Ovando to Vázquez-Ovando throught the main body of the manuscript. The reference of Wood and Lass is inaccurate; the citation needed is for the author(s) of the chapter that is being referenced.
In the manuscript, including Tables, the study material is sub-divided as nine populations and identified as ‘geographic population a priori’. However, groups 6-9 are reference trees and should not be treated as populations. This is especially so for the GenBank samples and the INIFAP samples, which represent germplasm that could have been collected from different geographic populations. The validity of using such artificial constructed groups as a priori geographic groups within the data analysis may skew the statistical output. Can the authors provide evidence that this will not be a concern for the algorithms of the software that was implemented?

Further, each ‘pop’ was broken down into Criollo, Non-Criollo and Admixture. However, the authors neglected to state the membership contribution based on Motamayor’s (2008) ten populations. Criollo was said to be high ancestry and instead the minimum inclusion level should have been stated. Similarly, which group(s) did the non-Criollo i.e. Forastero samples belong to and were these pure 100% individuals or was some minimum inclusion level applied e.g. 80% membership. For the admixtures, it would be of benefit to the reader to know which groups were represented by each accession. For example, what is the ancestral constitution of each sample? Since cacao germplasm collections, like other germplasm collections, contain mislabelled accessions, are the authors reasonably certain of the identity of the CATONGO, EET 399 [ECU], and RIM 24 [MEX]? Please also note the preferred accession name according to the ICGD database maintained by the University of Reading and available online.

Raw data was not presented or was not available.

Experimental design

This section was fairly well done. The authors are asked though why PCR products were sized on EtBr/PAGE system when a CEQ8000 system was available. The latter would have given more accurate size comparisons. The seven reference accessions from the NCBI database should be given the correct nomenclature using the preferred naming system found in the ICGD. Please note that Scavina is an accession group and not an accession like for example SCA 6, SCA 12 or SCA 24. Which of the Scavina accessions was actually used? Is the accession Matina-06 correct? Or is accession really MATINA 1/6?

Validity of the findings

Comments to the authors for this section cover results, discussion and conclusions.

A thorough evaluation of this section was hampered by the poor English. However, there were several areas that could be identified that needed to be addressed in order to improve the paper.
In the first paragraph of the results (lines 185-187), there is uncertainty of the total number of scored sites for genetic profiling. Are the six polymorphic sites comprised of a substitution event and five indel events? Are the six segregating sites different from the six polymorphic sites or are these the same? If the latter, then text could read instead ‘…of which only six segregating polymorphic sites (Table 2) were present as five indel events (Figure 1) and a substitution event.’

The presentation of the haplotypes including the network approach was well done barring (a) the colour scheme and font size in Figure 2 and (b) the possible clerical error of only 1 entry for population 8 for haplotype 11 in Table 2.

However, the authors could have presented the haplotypes for each accession, so that in each ‘population’ the reader could easily see which accessions had which haplotype. This information could then be discussed in light of other chloroplast haplotpyes from SSR or SNPs and known genetic ancestry of cacao. The publications of Kane et al. (2012), and Yang et al. (2011, 2013) become especially relevant. The latter two were not included in the manuscript though. Also, the haplotype diversity from the manuscript is only based on one area and the authors need to discuss whether a valid and acceptable network is possible using only one area.

The CRIOLLO 22 appeared different from the Soconusco Criollos. However, the reader is unsure which haplotype(s) are involved. Is CRIOLLO 22 represented by haplotype H11? Since chloroplast DNA is maternally inherited and all the Criollos have high membership, the presented research would suggest that more than one mother plant generated the Criollo accession group. Yang et al (2013) had 3 haplotypes based on SNP work of which only one was Criollo. More than one Criollo haplotype, if from members with >99% ancestry in the Criollo group is therefore a very intriguing finding.

The authors did not discuss their findings in light of known genetic groups or previous chloroplast haplotype studies. In Pop 9 with diverse members from at least three population groups – Criollo, Amelonado, Contamana- the results had the non-Criollo and admixed samples as having the same haplotype. It would of interest to cacao researchers to know if the MATINA had a Criollo haplotype or an AMELONADO haplotype as the provenance of this group is uncertain. Scavina belongs to the Contamana genetic group and Amelonado to the Amelonado genetic group. The results therefore indicate that these two different groups had a common maternal origin. This appears to be contrary to that presented in Yang et al. (2013).

Additional comments

This work needs to be published and made available to the cacao community. Please rewrite and resubmit. In particular, please correct for English usage and clarity.

---

## Round 0.2 · Minor Revisions

The reviewers were very positive in their opinions of your revised manuscript, but also provided detailed comments that I believe will further improve it. I also annotated a pdf of the manuscript where the presentation was unclear to me. Please consider all of our comments and suggestions. I look forward to receiving your revised manuscript.

·

Basic reporting

The English was revised and is clearer than the previous version. The background information is relevant to the work ad adequate.
Figure 2 is more clear than the original. But I think that the map is less clear than the original figure. To indicate the amplificationfrom Mexico to Chiapas to Soconusco, I think that is better to separate the image of Mexico and Chiapas (this could be in the right of the Mexico image), and use lines or arrows to indicate the amplification, similar to the original figure. As Soconusco image is separated, only indicate the amplification with lines or arrows.
In table 3 the author indicates row of Total for all the columns. But in the case of Hd ± sd and πd ± sd columns the value is the average, as indicate in Results, line 201. Should include another row for the average.
In table 4 indicates that Fst=0.0439, and in line 234 Fst=0.04339. In line 234 could eliminate “(Table 4)” because all the paragraph refers to this table and it is indicated at the end.
In line 245 indicate the number of insertions or deletions.

Experimental design

The research is rigorous, and the information included allow the replication of the research.

Validity of the findings

The research is relevant to the knowledge of the cultivation history of cacao, especially because was made on Mexico, one of the first countries with cultivated cacao.

Additional comments

The author made a good work including the recommendation of reviewers and editor.

·

Basic reporting

The revised manuscript is a much improved version and will be a good addition to the cacao literature. Suggestions and amendments are provided in the attached pdf.

Experimental design

Good reporting. Research area, question and knowledge gap meets journal policy.

Validity of the findings

Article meets journal standards. Discussion is well done based on the results presented.

Additional comments

Please see attached pdf for all comments for minor revisions if these are acceptable.
I look forward to seeing this article in the literature.

---

## Round 0.3 · accepted · Accept

I made a few small suggested corrections as I was reading the manuscript. See the attached file.